# Comparing Mindful and Non-Mindful Exercises on Alleviating Anxiety Symptoms: A Systematic Review and Meta-Analysis

**DOI:** 10.3390/ijerph17228692

**Published:** 2020-11-23

**Authors:** Wendy Wing Yan So, Erin Yiqing Lu, Wai Ming Cheung, Hector Wing Hong Tsang

**Affiliations:** 1Department of Rehabilitation Sciences, The Hong Kong Polytechnic University, Hung Hom, Hong Kong; wendy.so@connect.polyu.hk (W.W.Y.S.); erin.lu@polyu.edu.hk (E.Y.L.); 2Faculty of Education, The University of Hong Kong, Pokfulam, Hong Kong; cwming@hku.hk

**Keywords:** anxiety, mindful exercise, physical exercise, primary health care, yoga, qigong

## Abstract

Background: In recent years, studies and reviews have reported the therapeutic benefits of both mindful and non-mindful exercises in reducing anxiety. However, there have not been any systematic reviews to compare their relative effectiveness for therapeutic application, especially among the non-clinical population. Thus, the aim of this review is to compare the effectiveness between mindful and non-mindful exercise on treating anxiety among non-clinical samples. Methods: Potential articles were retrieved from PubMed, Embase, Academic Search Premier, and PsycInfo. Randomized controlled trials, which involved both mindful and non-mindful exercises as intervention, and the use of anxiety outcome measures were included. Results: Twenty-four studies fulfilled the inclusion criteria and were included in our systematic review. In addition, 14 studies provided sufficient data to be included in the meta-analysis. For studies that reported significant group differences at post-assessment, results showed that mindful exercise was more beneficial in reducing anxiety than non-mindful exercise. The meta-analysis reported that yoga was more effective in reducing anxiety than non-mindful exercise. Conclusions: Compared to non-mindful exercise, yoga is shown to be more effective in alleviating anxiety symptoms. It is recommended that yoga could be used as a primary healthcare intervention to help the public reduce anxiety.

## 1. Introduction

Anxiety refers to a kind of psychological state that is characterized by apprehensive expectation or fear [1]. It usually arises when facing stressful events [2]. Although it is a normal response, once the feeling of being anxious becomes overwhelming, it may lead to the development of anxiety disorders. Anxiety disorder is one of the most common psychiatric conditions [3]. This review focuses on the treatment of anxiety, not anxiety disorder.

Currently, the most common treatment for anxiety disorders is psychopharmacology [4]. However, this is often associated with side effects and other problems, including metabolic abnormalities, dependency, withdrawal, etc. [5]. Other forms of commonly used treatment include psychotherapy and cognitive behavioral therapy. Both types of treatment are time-consuming and manpower intensive, and also require intervention from qualified health professionals, such as clinical psychologists and occupational therapists that many clients cannot afford. Moreover, a study showed that cognitive behavioral therapy would be most beneficial when conducted individually [6], which adds to its high cost. Therefore, the most effective way is to cope well with the daily anxiety we may experience to prevent oneself from being diagnosed with anxiety disorders.

Physical exercise, which is already well-known to be beneficial to people’s physical health, has recently been used to alleviate anxiety in the non-clinical population. Shirifard, et al. [7] found that, compared to the control group, the anxiety level of the participants who joined a physical exercise program decreased significantly. Another study also suggested that physical exercise may serve as an effective treatment for anxiety disorders [8]. A review showed that physical exercise might be helpful for both clinical and non-clinical samples of anxiety [3]. 

In recent years, a growing number of research studies have been focusing on the effect of a special type of exercise—mindful exercise—on anxiety. Mindful exercise refers to physical exercise with a mental emphasis at the same time [9]. It usually involves low to moderate physical movement with a mental focus on breathing and meditation at the same time [10]. Yoga, tai chi, and qigong are popular mindful exercises. A recent systematic review suggested that yoga might be effective in serving as a safe intervention for non-clinical conditions with anxiety problems [4]. In addition, a review on the effect of qigong on anxiety yielded similar results. Yin and Dishman [11] found that qigong had favorable effects on reducing anxiety symptoms.

Based on the above literature review, both mindful and non-mindful (or physical) exercises are likely to be useful in alleviating anxiety. However, it remains uncertain if available evidence provides sufficient support for regular therapeutic application. It remains unknown which of these interventions (mindful vs. non-mindful) is superior to the other in alleviating anxiety symptoms in non-clinical and/or clinical conditions.

The aim of this review is therefore to compare the relative effectiveness of two exercise interventions, namely mindful and non-mindful exercises, on reducing or treating anxiety as a psychological state in the non-clinical population. Existing reviews compared the differential effects of mindful and non-mindful interventions on depression [10] and schizophrenia [9]. Tsang, Chan and Cheung [10] did not find any difference between mindful and non-mindful exercises in reducing depressive symptoms. On the other hand, Li, Shen, Wu, Tan, Sun, Keller, Jiang and Wu [9] found that mindful intervention, compared to non-mindful intervention, is effective in reducing psychiatric symptoms. However, the number of studies included in the review was limited. To date, no systematic reviews or meta-analyses have ever been conducted on comparing the effects of mindful and non-mindful exercises on treating anxiety. Since the prevalence of symptoms of anxiety in the general public has had a rising trend [12], it is timely to evaluate if these two non-pharmacological interventions are effective, and identify which of them offers more benefits to people suffering from anxiety.

## 2. Methods

Our review followed the Preferred Reporting Items for Systematic Reviews and Meta-Analyses (PRISMA) guidelines.

### 2.1. Search Strategy

Studies were identified through a systematic search of relevant electronic databases, including PubMed, Embase, Academic Search Premier, and PsycInfo, from inception to November 2019 using the following search terms: (anxiety) AND (yoga OR “tai chi” OR “tai Ji” OR qigong OR “qi gong” OR “mindful exercise” OR “mind body exercise”) AND (aerobic OR “physical exercise” OR “conventional exercise” OR walk * OR jog * OR run * OR cycl * OR swim * OR anaerobic OR danc * OR stretch * OR “non mindful”) AND (randomized controlled trial). No time limit was set. Relevant systematic reviews and reference lists of selected studies were screened to identify additional studies.

### 2.2. Inclusion and Exclusion Criteria

Journal articles that fulfilled the following criteria were included: (A) being a randomized controlled trial (RCT); (B) journal articles; (C) including mindful exercise and non-mindful exercise as intervention; (D) including an anxiety measurement as the outcome; (E) including participants who had an identified type of anxiety or healthy adults with elevated levels of anxiety at the commencement of the RCT, measured by a validated clinician-based or self-reported anxiety symptom questionnaire but without a formal diagnosis of an anxiety disorder; and (F) published in Chinese or English. Studies that combined intervention of mindful or non-mindful nature with other types of intervention or studies in which participants were diagnosed with multiple psychological disorders were excluded.

### 2.3. Study Selection

Two independent reviewers with a degree in psychology screened all abstracts and titles under the supervision of the corresponding author. Studies that had at least one reviewer considered as eligible were included for the next tier of screening. After that, the full texts of the included papers were downloaded and evaluated following the inclusion and exclusion criteria by the two reviewers. Disagreement was solved by discussion and reconciliation; the final decision was mutually agreed upon with mediation by the corresponding author.

### 2.4. Data Extraction

One reviewer extracted the data from each of the selected articles, while the other crosschecked the information. Information on the study’s authors and year, sample size, sample characteristics, type, frequency and duration of intervention, and anxiety-related outcomes were extracted.

### 2.5. Data Synthesis

Since different studies used different assessment tools to measure anxiety, standardized mean difference was used instead of mean differences to calculate the pooled effect size. Standardized mean differences and their 95% confidence intervals were calculated using Review Manager 5.3. (Nordic Cochrane Centre, Cochrane Collaboration, 2014 Copenhagen, Denmark) Heterogeneity was assessed for each meta-analysis by looking at the *I*^2^ value. A funnel plot and Egger’s regression were examined to detect publication bias using Comprehensive Meta-Analysis version 3. Significant result in Egger’s regression (*p* < 0.05) indicated that publication bias may have existed. Studies that did not provide enough information to calculate effect size were described in qualitative means.

### 2.6. Quality Assessment of Studies

Each included study was analyzed for risk of bias using the Cochrane Collaboration’s criteria [13]. Two reviewers conducted the analysis separately and results were then compared. Disagreement was solved by discussion.

## 3. Results

### 3.1. Search Results

A total of 1453 articles were retrieved from the electronic databases. Reference lists of included studies and related systematic reviews were screened, and an additional 18 studies were identified. After removing duplication, 1284 titles and abstracts were screened. Further screening was conducted for 147 articles by reading the full texts. The main reasons for exclusion were absence of mindful or non-mindful intervention, absence of anxiety measures, and using research methodologies other than RCT. Finally, 24 studies fulfilled the inclusion criteria and were included in our systematic review. Unfortunately, the data needed for meta-analysis were not available for 10 studies. As a result, 14 studies were included in the meta-analysis. The search results are summarized in the PRISMA flow diagram (Figure 1).

### 3.2. Quality Assessments

The risk of bias of the included studies is summarized in Figure 2. Due to the nature of the study, all trials were rated as high risk in performance bias, since blinding the participants was difficult. Three studies showed low risk of bias in all other categories [14,15,16]. The remaining studies showed high risk of bias because the randomization process, blinding procedure, and reasons of attrition were not reported with sufficient detail. For the inclusion criteria of some trials, the experience of practicing mindful or non-mindful exercise was not specified, which may have also created bias.

### 3.3. Characteristics of Included Studies and Participants

The year of publication of the included studies was between 1991 to 2019. Eleven studies were conducted in the USA [16,17,18,19,20,21,22,23,24,25,26], three studies in India [27,28,29], three studies in Hong Kong [15,30,31], two studies in Brazil [32,33], one study in Sweden [34], one study in Iran [35], one study in China [36], one study in Australia [14], and one study in Canada [37].

Sample size ranged from 21 to 200. Participants’ age ranged from 13 to over 90 years old. While ten studies involved healthy participants [19,20,21,22,23,24,27,29,33,36], the rest of the studies involved a clinical population that included patients with cancer [15,25,37], chronic pain [17,28], fibromyalgia [26], heart failure [34], multiple sclerosis [16,35], hypertension [31], insomnia [32], knee osteoarthritis [14,18], and Parkinson’s disease [30].

For anxiety outcome measurements, various scales were used in the included studies. Nine studies used the State-Trait Anxiety Inventory [16,17,19,21,22,23,24,28,33], eight studies used the Hospital Anxiety and Depression Scale [15,18,20,26,27,30,34,37], two studies used the Beck Anxiety Inventory [31,32], one study used the Hamilton Anxiety Rating Scale [29], one study used the Global Severity Index [25], and one study used the Piers–Harris Children’s Self-Concept Scale [36]. All of the above information is summarized in Table 1.

### 3.4. Intervention Characteristics

All included studies used either yoga or qigong as mindful intervention (Table 1). Sixteen studies used yoga as intervention [16,17,18,19,20,21,22,24,27,28,29,30,32,33,34,35], while eight studies used qigong. Among the studies that used qigong, five studies adopted tai chi [14,23,25,26,36] and three studies adopted Guolin qigong [15,30,31,37]. Common non-mindful exercises used in the included studies were walking, aerobic exercise, and stretching. The duration of intervention ranged from seven days to 24 weeks. The intensity of intervention varied across different studies, from practicing once a week [16,18,24,30], twice per week [14,15,17,25,26,31,32,33,34,37], three times a week [19,20,21,22,27,35], five times a week [23,36], to daily practice [28,29]. In general, most of the studies required participants to practice twice or three times per week for 60 min.

### 3.5. Summary of Findings

Compared to the baseline, five studies showed a significant decrease in anxiety in the mindful exercise group at post-assessment [22,27,28,29,32]. In addition, Liu, You, Loo, Sun, He, Sit, Jia, Wong, Xia, Zheng, Wang, Wang, Lao, Chen and Loo [15] found a significant reduction in anxiety at follow-up assessment, which was conducted 24 weeks after the intervention, compared to the baseline. Two studies found a significant drop in anxiety in the non-mindful exercise group at post-assessment [14,17]. On the other hand, Satyapriya, Nagarathna, Padmalatha and Nagendra [27] found a significant increase in anxiety for non-mindful exercise group at post-assessment.

Five studies reported that both mindful and non-mindful exercise groups showed a significant decrease in anxiety at post-assessment [20,21,23,34,35]. On the other hand, during post-assessment, eight studies showed significantly lower anxiety levels in the mindful exercise group compared to non-mindful exercise group [18,22,25,27,28,30,35,36]. Last but not least, three studies found a significant interaction effect which showed lower anxiety level in mindful exercise group at post-assessment [24,28,30]. Furthermore, Bonura and Tenenbaum [24] found that this significant interaction effect lasted for one month after intervention.

### 3.6. Effectiveness of Intervention (Meta-Analysis)

Ten studies did not provide sufficient data to calculate the standardized mean differences. As a result, 14 studies were eligible for inclusion in the meta-analysis (Table 2). Ten studies used yoga as mindful exercise [17,18,21,27,28,29,30,32,34,35], while four studies used qigong as mindful exercise [14,23,36,37]. Overall, there was no significant difference in the level of anxiety between practicing mindful and non-mindful exercise (SMD = −0.23 (95% CI = −0.58 to 0.11), *p* = 0.18, *I*^2^ = 86%; Figure 3, Table 3). Further meta-analyses were conducted to compare the differences between yoga and non-mindful exercise as well as qigong and non-mindful exercise in reducing anxiety. When comparing yoga with non-mindful exercise, a significant difference was found, which indicated that yoga was more effective in reducing anxiety (SMD = −0.45 (95% CI = −0.81 to −0.09), *p* = 0.01, *I*^2^ = 82%; Figure 4). On the other hand, when comparing qigong with non-mindful exercise, no significant difference was found (SMD = −0.04 (95% CI = −0.43 to 0.35), *p* = 0.85, *I*^2^ = 62%; Figure 5, Table 4). Egger’s regression tests were conducted, and no evidence of publication bias or asymmetry in the funnel plot was found in any of the meta-analyses conducted.

## 4. Discussion

To our knowledge, this is the first meta-analytical study that compared the effect of mindful and non-mindful exercises on anxiety in a non-clinical population. The aim of this study was to compare the effects of mindful and non-mindful exercises on anxiety. Twenty-four studies were included in the qualitative synthesis and 14 studies were included in the meta-analysis. Heterogeneity was observed, since the type of participants, duration of intervention, and assessment tools for anxiety were different across studies. While some included studies showed that both mindful and non-mindful interventions had a significant effect on the reduction of anxiety, eight studies found more significant reduction in anxiety after mindful intervention compared to non-mindful intervention. Three studies indicated a significant interaction effect for the mindful exercise group at post-assessment. Although the results of the meta-analysis showed that there were no differences between mindful and non-mindful interventions on anxiety, when qigong and yoga was separately analyzed, significant results were found. When comparing yoga with non-mindful intervention, a medium and significant effect was found; participants who practiced yoga demonstrated a lower anxiety level after intervention. In addition, when comparing qigong and non-mindful intervention, no significant effect was identified. It is important to note that only four studies were included in the meta-analysis of qigong and non-mindful exercise. This may not provide enough evidence for us to draw any valid conclusions. Nonetheless, it is possible that even though yoga and qigong are both mindful exercises, they may not produce the same effect on anxiety.

A possible explanation as to why yoga is more effective in reducing anxiety than non-mindful exercise is that it has incorporated the regulation of breathing as well as relaxation exercises in the practice [18]. This is the uniqueness of mindful practice compared to non-mindful exercise, with mindful exercise focusing on both the mind and the body [15]. Himashree, Mohan and Singh [29] suggested that yoga may result in a decrease in the sympathetic discharge and better oxygen saturation. They suggested that these could be the potential reasons that lead to improvement of overall mental fitness and health. Another possible mechanism is the reduction of HPA activity, which reduces sympathetic arousal and results in stability in the autonomic system [27]. However, this cannot explain why qigong is not as effective as yoga, because it also involves regulation of breathing and relaxation in its practice.

One possible mechanism explained by Bonura and Tenenbaum [24] is that yoga helps reduce anxiety through one more pathway, which involves the improvement of self-control. The assessment of self-control measured the tendencies to apply self-control methods as a solution to behavioral problems [38]. This group of researchers found an increase in self-control in the yoga group, and their regression result suggested that changes in self-control may be predictive of changes in psychological health. Another study that investigated the mechanisms of yoga in reducing symptoms of post-traumatic stress disorder found that yoga is useful in reducing expressive suppression. This may be due to the emphasis on a nonjudgmental attitude toward thoughts and experiences throughout the practice of yoga. It also helps people reduce their efforts in coping with distressing emotions [14]. On the other hand, no studies have been done to provide evidence that qigong could enhance self-control or decrease expressive suppression. Moreover, the type of qigong selected may not be suitable for the population of this study. Bao and Jin [36] suggested that the type of qigong they used may be too complicated for adolescents. Thus, the participants may only focus on memorizing the posture, which hinders the beneficial effects of qigong. Another study used tai chi as an intervention for patients with knee osteoarthritis [14]. Since practicing tai chi requires lots of bending of the knees, the participants may experience knee pain throughout the practice. Another possible reason is that, among the four studies which were included in the meta-analysis, three studies were conducted in western countries, and the qualification of the instructor was not mentioned. This may largely affect the findings of the studies, as the effectiveness of qigong in these studies is questionable. Furthermore, since qigong was originally developed in China, it may not be as effective as yoga when it is practiced by foreigners due to the factor of cultural compatibility. An increasing popularity of yoga was observed [39]. The lifetime prevalence of yoga was 13.2%, while the lifetime prevalence of qigong was only 3.1% in the United States [40]. This shows that cultural compatibility may have an important role in determining whether a certain mindful exercise is more effective than the other in a different context. However, the mechanism that explains why the cultural factor may lead to this difference remains unknown. Further studies are needed to explore further in this aspect.

### Limitations and Future Directions

Several limitations could be identified for this review. First, the number of studies included in the meta-analysis of qigong and non-mindful exercises was limited. The results of this review should be interpreted with caution. More RCTs involving a larger sample should be conducted to compare the differential effects of qigong, yoga, and non-mindful exercise on anxiety. The results may have been different with more RCTs available for comparison. Second, the included studies employed a diversity of population, and the intensity of interventions varied widely. This may not provide sufficient evidence to recommend which population and intensity could be most beneficial or therapeutic for reducing anxiety. The age and diagnosis of the participants should also be considered carefully by the researchers before deciding which mindful intervention will be used. Although all of the included studies were RCTs, almost half of the included studies had selection bias, detection bias, or attrition bias. Future studies related to this line of research should try to minimize these biases in their study designs. Furthermore, more research studies with a higher quality are needed, especially in clinical samples with anxiety as the primary complaint and clinical disorder. Last but not least, since this review presented evidence that yoga could be more beneficial in reducing anxiety than non-mindful exercise, future research could begin to investigate the physiological and psychological mechanisms that explain why yoga is superior to non-mindful exercise in reducing anxiety.

## 5. Conclusions

To conclude, the results of the meta-analysis that compared the effectiveness of yoga and non-mindful exercise show that yoga is more beneficial in alleviating anxiety symptoms than non-mindful exercise. However, qigong is not supported by this review to be more advantageous than non-mindful exercise in treating anxiety. More studies are required to identify the reasons behind this. This review provides sufficient evidence to suggest that yoga may be used as a regular intervention in primary healthcare and clinical settings in helping people who practice it have beneficial effects on relieving anxiety symptoms. Further studies should explore the underlying physiological and psychological mechanism that underpins its clinical effectiveness to turn it to the mainstream instead of adjunct intervention for people with anxiety.

## Figures and Tables

**Figure 1 ijerph-17-08692-f001:**
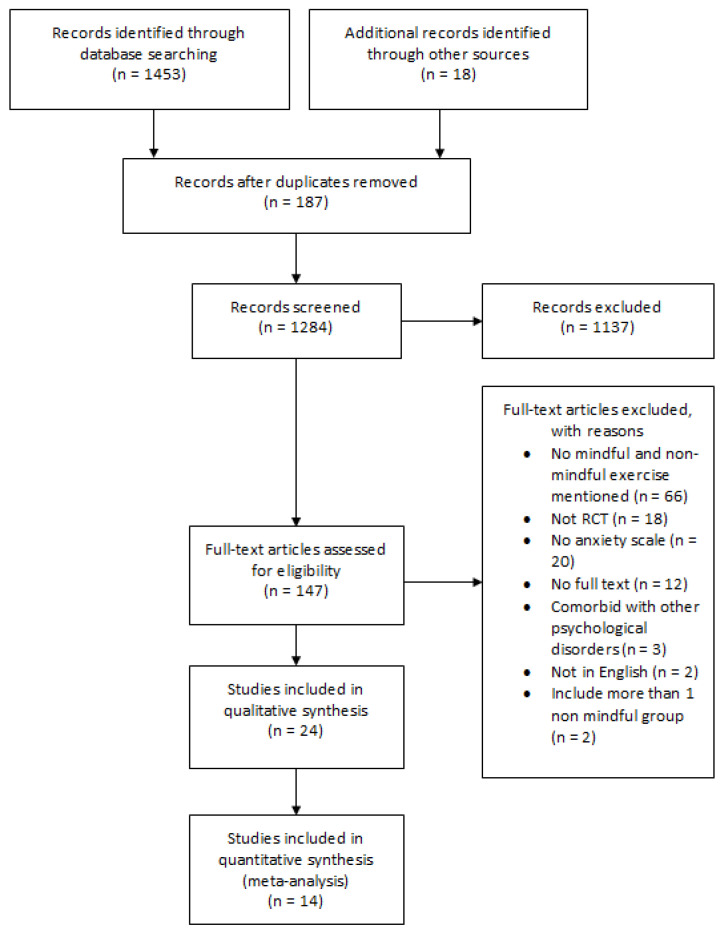
Flow diagram of the study selection.

**Figure 2 ijerph-17-08692-f002:**
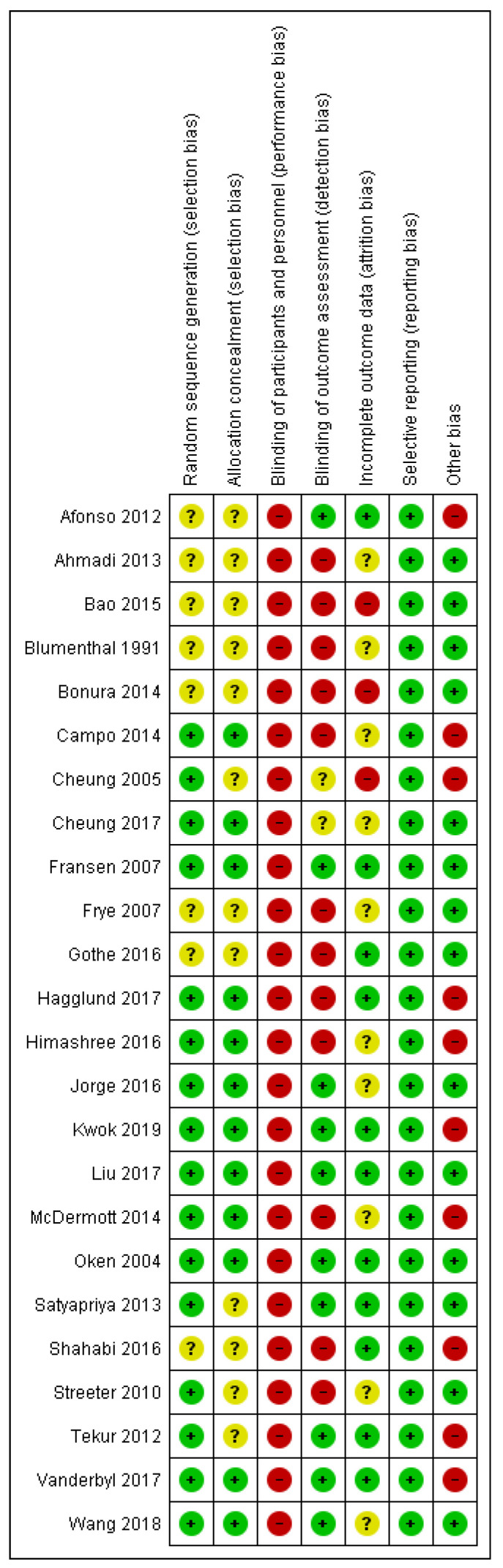
Summary of risk of bias.

**Figure 3 ijerph-17-08692-f003:**
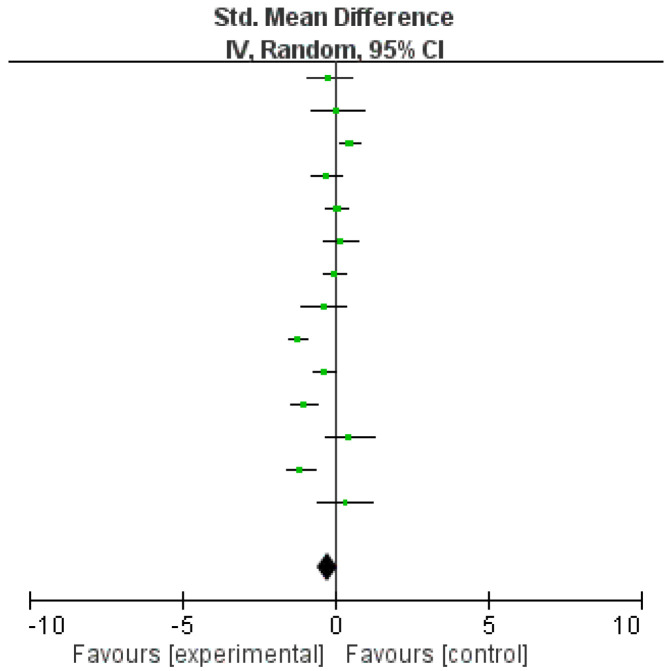
Effect of mindful exercise on anxiety symptoms in comparison to non-mindful exercise.

**Figure 4 ijerph-17-08692-f004:**
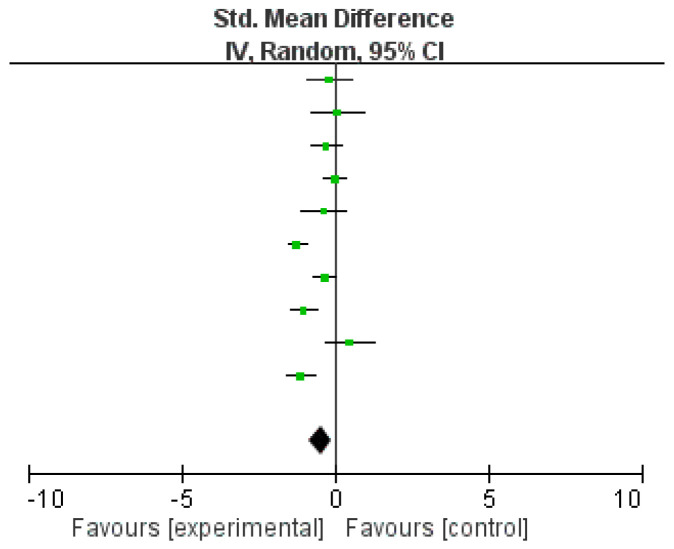
Effect of yoga on anxiety symptoms in comparison to non-mindful exercise.

**Figure 5 ijerph-17-08692-f005:**
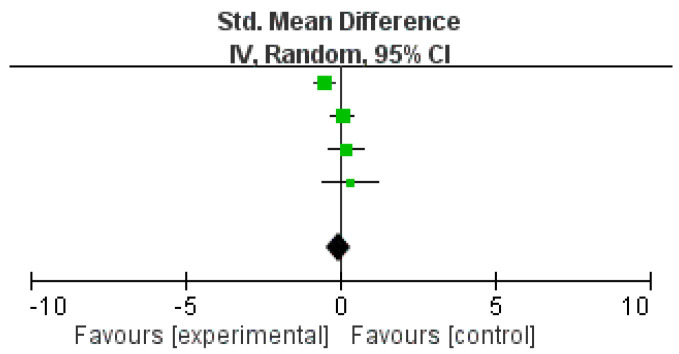
Effect of qigong on anxiety symptoms in comparison to non-mindful exercise.

**Table 1 ijerph-17-08692-t001:** Summary of the included studies.

Study	Age and Gender	Sample Size	Health Status	Type of Mindful Exercise	Type of Non-Mindful Exercise	Duration	Outcomes
Afonso et al., (2012) [32]	Age: 50−65 years Gender: Female	29	Postmenopausal with Insomnia	Yoga	Stretching	four months 60 min, twice per week	BAI
Ahmadi et al., (2013) [35]	Age: 22−54 years Gender: 0M, 21F	21	Multiple sclerosis	Yoga	Treadmill training	eight weeks 60 min, three times per week	BAI
Bao et al., (2015) [36]	Age: 13−16 years Gender: 75M, 85F	160	Healthy	Tai chi	Gymnastics	one year 60 min, five times a week	PHCSCS
Blumenthal et al., (1991) [19]	Age: 60−83 years Gender: not specify	67	Healthy	Yoga	Aerobic	four months 60 min, three times per week	STAI
Bonura et al., (2014) [24]	Age: 65−92 years Gender: 32M, 74F	106	Healthy	Chair yoga	Chair exercise	six weeks 45 min, once per week	STAI
Campo et al., (2014) [25]	Age: 58−93 years Gender: 29M, 0F	29	Senior prostate cancer survivors	Qigong	Stretching	12 weeks 60 min, twice per week	GSI
Cheung et al., (2005) [31]	Age: 18−75 years Gender: 37M, 51F	88	Hypertension	Qigong	Conventional exercise	four weeks 120 min, twice per week	BAI
Cheung et al., (2017) [18]	Age: ≥ 60 years Gender: Female	83	Knee osteoarthritis	Yoga	Aerobic/strengthening exercises	eight weeks 45 min, once per week	HADS
Fransen et al., (2007) [14]	Age: 59−85 years Gender: 33M, 78F	111	Osteoarthritis	Tai chi	Hydrotherapy	12 weeks 60 min, twice per week	DASS21
Frye et al., (2007) [23]	Age: ≥ 50 years Gender: not specify	61	Healthy	Tai chi	Low impact exercise	12 weeks, 60 min, five times per week	STAI
Gothe et al., (2016) [21]	Age: 55−79 years Gender: 26M, 92F	118	Healthy	Yoga	Stretching and strengthening	eight weeks 60 min, three times per week	STAI
Hägglund et al., (2017) [34]	Age: 18−80 years Gender: 26M, 14F	40	Heart failure	Yoga	Hydrotherapy	12 weeks 60 min, twice per week	HADS
Himashree et al., (2016) [29]	Age: 20−30 years Gender: 200M, 0F	200	Healthy	Yoga	Physical exercise (jogging, squats, sit-ups, etc.)	60 days 1 h, daily	Hamilton’s Anxiety Scale
Jorge et al., (2016) [33]	Age: 45−65 years Gender: not specified	69	Healthy	Yoga	Physical exercise	12 weeks 75 min, twice per week	STAI
Kwok et al., (2019) [30]	Age: not specified Gender: 65M, 73F	138	Parkinson’s Disease	Yoga	Stretching and resistance exercise	eight weeks Yoga: 90 min, once per week Stretching and resistance exercise: 60 min, once per week	HADS
Liu et al., (2017) [15]	Age: 21−80 years Gender: 0M, 158F	158	Breast Cancer	Qigong	Physical stretching	24 weeks 60 min, twice per week	HADS
McDermott et al., (2014) [20]	Age: 30−65 years Gender: 16M, 25F	41	Healthy (individuals with a first-degree relative with T2DM)	Yoga	Walking	eight weeks Yoga: At least three times per week, 75 min Walking: six days per week, 30 min plus breaks for rest	HADS
Oken et al., (2004) [16]	Age: not specified Gender: 4M, 33F	37	Multiple sclerosis	Yoga	Aerobic exercise	six months 90 min, once per week	STAI
Satyapriya et al., (2013) [27]	Age: 20−35 years Gender: 0M, 96F	96	Pregnant	Yoga	Standard antenatal practices	one month 120 min, three times per week	HADS STAI
Shahabi et al., (2016) [17]	Age: 18−65 years Gender: 3M, 24F	27	Chronic abdominal pain or discomfort and associated bowel habit changes	Yoga	Walking	16 sessions 60 min, biweekly	STAI
Streeter et al., (2010) [22]	Age: 18−45 years Gender: 12M, 22F	34	Healthy	Yoga	Walking	12 weeks 60 min, three times per week	STAI
Tekur et al., (2012) [28]	Age: 18−60 years Gender: 44M, 36F	80	Chronic low back pain	Yoga	Physical exercise	seven days whole day, everyday	STAI
Vanderbyl et al. (2017) [37]	Age: ≥ 18 years Gender: 14M, 10F	24	Advanced NSCLC or GI cancer	Qigong	Standard endurance and strength training	six weeks 45 min, twice per week	HADS
Wang et al., (2018) [26]	Age: ≥ 21 years Gender: 3M, 108F	111	Fibromyalgia	Tai chi	Aerobic exercise	24 weeks 60 min, twice per week	HADS

**Table 2 ijerph-17-08692-t002:** Effect of mindful exercise on anxiety symptoms in comparison to non-mindful exercise.

	Mindful Exercise			Non Mindful Exercise				Std. Mean Difference
Study or Subgroup	Mean	SD	Total	Mean	SD	Total	Weight	IV. Random, CI 95%
Afonso 2012 [32]	8.8	7.359	15	10.2	7.109	14	6.3%	−0.19 (−0.92, 0.54)
Ahmadi 2013 [35]	6.45	3.61	11	6.1	4.95	10	5.7%	0.08 (−0.78, 0.93)
Bao 2015 [36]	6.51	2.19	73	5.28	2.9	69	8.1%	0.48 (0.14, 0.81)
Cheung 2017 [18]	3.8	4.993	32	5.2	5.158	28	7.4%	−0.27 (−0.78, 0.24)
Fransen 2007 [14]	5.1	6	56	4.6	5.2	55	8.0%	0.09 (−0.28, 0.46)
Frye 2007 [23]	47.1	9.66	23	45.3	8.7	28	7.2%	0.19 (−0.36, 0.75)
Gothe 2016 [21]	30.71	9.7	58	30.8	9.7	50	8.0%	−0.01 (−0.39, 0.37)
Hagglund 2017 [34]	2.7	3.2	18	3.9	3.4	12	6.3%	−0.36 (−1.09, 0.38)
Himashree 2016 [29]	8.86	4.3	100	14.31	4.6	100	8.2%	−1.22 (−1.52, −0.92)
Kwok 2019 [30]	3.97	3.57	71	5.22	3.84	67	8.1%	−0.34 (−0.67, 0.00)
Satyapriya 2013 [27]	5.22	1.36	51	7.82	3.43	45	7.8%	−1.01 (−1.44, −0.59)
Shahabai 2016 [17]	53.9	10.9	17	49.3	5.4	10	6.0%	0.48 (−0.31, 1.27)
Tekur 2012 [28]	33.43	8.08	40	43.68	9.89	40	7.5%	−1.12 (−1.60, −0.65)
Vanderbyl 2017 [37]	5.5	2.1	10	4.5	3.3	9	5.4%	0.35 (−0.56, 1.26)
**Total (95% CI)**			**575**			**537**	**100.0%**	**−0.23 (−0.58, 0.11)**

Heterogenity: Tau^2^ = 0.35; Chi^2^ = 94.66, df = 13 (*p* < 0.00001); *I*^2^ = 86%. Test for overall effect: Z = 1.34 (*p* = 0.18).

**Table 3 ijerph-17-08692-t003:** Effect of yoga on anxiety symptoms in comparison to non-mindful exercise.

	Yoga			Non Mindful Exercise				Std. Mean Difference
Study or Subgroup	Mean	SD	Total	Mean	SD	Total	Weight	IV. Random, CI 95%
Afonso 2012 [32]	8.8	7.359	15	10.2	7.109	14	8.5%	−0.19 (−0.92, 0.54)
Ahmadi 2013 [35]	6.45	3.61	11	6.1	4.95	10	7.5%	0.08 (−0.78, 0.93)
Cheung 2017 [18]	3.8	4.993	32	5.2	5.158	28	10.4%	−0.27 (−0.78, 0.24)
Gothe 2016 [21]	30.71	9.7	58	30.8	9.7	50	11.5%	−0.01 (−0.39, 0.37)
Hagglund 2017 [34]	2.7	3.2	18	3.9	3.4	12	8.5%	−0.36 (−1.09, 0.38)
Himashree 2016 [29]	8.86	4.3	100	14.31	4.6	100	12%	−1.22 (−1.52, −0.92)
Kwok 2019 [30]	3.97	3.57	71	5.22	3.84	67	11.8%	−0.34 (−0.67, 0.00)
Satyapriya 2013 [27]	5.22	1.36	51	7.82	3.43	45	11.1%	−1.01 (−1.44, −0.59)
Shahabai 2016 [17]	53.9	10.9	17	49.3	5.4	10	8.0%	0.48 (−0.31, 1.27)
Tekur 2012 [28]	33.43	8.08	40	43.68	9.89	40	10.7%	−1.12 (−1.60, −0.65)
**Total (95% CI)**			**413**			**376**	**100%**	**−0.45 (−0.81, −0.09)**

Heterogenity: Tau^2^ = 0.26; Chi^2^ = 48.92, df = 9 (*p* < 0.00001); *I*^2^ = 82%. Test for overall effect: Z = 2.45 (*p* = 0.01).

**Table 4 ijerph-17-08692-t004:** Effect of qigong on anxiety symptoms in comparison to non-mindful exercise.

	Qigong			Non Mindful Exercise				Std. Mean Difference
Study or Subgroup	Mean	SD	Total	Mean	SD	Total	Weight	IV. Random, CI 95%
Bao 2015 [36]	−6.51	2.19	73	−5.28	2.9	69	32.8%	−0.48 (−0.81, −0.14)
Fransen 2007 [14]	5.1	6	56	4.6	5.2	55	31.0%	0.09 (−0.28, 0.46)
Frye 2007 [23]	47.1	9.66	23	45.3	8.7	28	23.2%	0.19 (−0.36, 0.75)
Vanderbyl 2017 [37]	5.5	2.1	10	4.5	3.3	9	13.0%	0.35 (−0.56, 1.26)
**Total (95% CI)**			**162**			**161**	**100%**	**−0.04 (−0.43, 0.35)**

Heterogenity: Tau^2^ = 0.09; Chi^2^ = 7.88, df = 3 (*p* < 0.05); *I*^2^ = 62%. Test for overall effect: Z = 0.20 (*p* = 0.85).

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
