# Peer review of "Comparing Mindful and Non-Mindful Exercises on Alleviating Anxiety Symptoms: A Systematic Review and Meta-Analysis"

_ijerph, 2020, doi:10.3390/ijerph17228692_

Round 1

Reviewer 1 Report

General comments

This article presents the results of a systematic review and meta-analysis of the effects of mindfulness and non-mindfulness exercise on reduced anxiety. As physical activity has been shown to be effective in reducing anxiety, the authors aimed at identifying whether mindful exercise (yoga, tai chi, qigong) would be more effective than non-mindful exercise in non-clinical populations. The results showed that yoga was more effective than on-mindful exercise.

The introduction is sufficiently documented, the methodology is sound, the conclusions are in line with the results, Tables and Figures are clearly presented and bring useful information, and the limitations are identified. This article can bring useful information for the field as currently the most common treatment for anxiety is medication, which can have side effects, and CBT, which is more effective in individual therapy. More easily accessible methods that can be used by a greater number of individuals without negative side effects is an interesting avenue for future research and practice.

A few suggestions are made in order to develop some arguments about the psychological processes which could explain the greater efficacy of yoga exercise compared to other mindful and non-mindful exercises.

Detailed comments

There are a few typos: chose the plural or add “a”,

for example L40 “a study”

L47 “Another study”

L191 “compared to”

L260-261 the sentence is unclear: is it methods “as a” solution?

L265 “population of this study” rather than study population

P12 In terms of content, it would be useful to add more about psychological processes to explain why yoga might be more effective that other mindful exercise or non-mindful exercise. For example self-compassion, psychological flexibility, non-judgment. These processes may be more stressed in yoga practice than in Taichi or Qicong, and more than in non-mindful exercise.

Furthermore, in the introduction it would be useful to explain why the authors chose not to include standard mindfulness programs as they also use mindful exercise.

Author Response

There are a few typos: chose the plural or add “a”,

for example L40 “a study”

Reply: It was revised; please see p. 2, line 40.

L47 “Another study”

Reply: It was revised; please see p. 2, line 48.

L191 “compared to”

Reply: It was revised; please see p. 6, line 194.

L260-261 the sentence is unclear: is it methods “as a” solution?

Reply: It was revised; please see p. 12, line 261.

L265 “population of this study” rather than study population

Reply: It was revised; please see p. 13, line 270.

P12 In terms of content, it would be useful to add more about psychological processes to explain why yoga might be more effective that other mindful exercise or non-mindful exercise. For example self-compassion, psychological flexibility, non-judgment. These processes may be more stressed in yoga practice than in Taichi or Qicong, and more than in non-mindful exercise.

Reply: Thank you for your comments. This information was added on p. 12, lines 264-268.

Furthermore, in the introduction it would be useful to explain why the authors chose not to include standard mindfulness programs as they also use mindful exercise.

Reply: The wordings have been revised to “mindful” and “non-mindful” (instead of “mindfulness” and “non-mindfulness”) to reduce confusion. Also, p. 2, lines 65-66 have been revised to make the aim clearer.

Reviewer 2 Report

Overall, the manuscript is clearly written and presents an interesting analysis. Please also see my more detailed comments below:

  • Abstract and title: The phrase “mindfulness exercise” is ambiguous, as it could simply mean practicing mindfulness, and it doesn’t necessarily imply that there is a physical exercise component attached with it. The term “non-mindfulness exercise” is then even more ambiguous, as it only says that there is some form of unspecified exercise but one not involving mindfulness. I suggest adding “mindfulness” and “non-mindfulness” to the phrase “physical exercise”, to make this clearer. Alternatively, say “physical exercise involving practices of mindfulness”.
  • Abstract: It is not clear until the final sub-section of the abstract what the authors mean about mindfulness exercise. And it is only indirectly defined. So, yoga seems to be considered a mindfulness exercise. What are others? Again, as above, please be clearer about the terms and definitions used. Note that both Li et al. (2018) and Tsang et al. (2008) use “mindful exercise” rather than “mindfulness exercise”, which offers a further alternative.
  • In Line 34, the authors state that the review focuses on anxiety disorders and not anxiety. However, in Line 35, anxiety is again used on its own. So, is anxiety used as a synonym for anxiety disorders here?
  • Line 40: grammar (“study showed”)
  • The paragraph starting with Line 51 addresses my concern above. However, it is a bit rushed. Most of the literature on mindfulness relates to practices without an exercise component and thus mainly sitting meditation. This needs to be explained first before mentioning mindfulness in the context of physical exercises. Otherwise uninformed readers link the vast mindfulness literature directly to physical exercise without appreciating this distinction. The present study focuses on a relatively small part of the mindfulness literature. However, this is not a criticism but something that can be seen as a strength.
  • The introduction is a bit rushed. What did Li et al. (2018) and Tsang et al. (2008) find in their reviews? Schizophrenia is quite distinct, but anxiety is very related to depression, which means that similar results might be expected compared to Tsang et al.
  • Further to my point that anxiety and depression are related: Quite often, interventions measure both outcomes in their studies. So, to what extent did studies included in Tsang et al. also measure anxiety? And how many studies found in your review were also included in Tsang et al.? Any overlap is important to note – not in the introduction of course, but at some stage in the manuscript.
  • Please mention and discuss effect sizes for these studies.
  • Discuss the potential of self-selection bias in the studies. Perhaps people opting into yoga interventions are somehow different to participanrts in qigong classes?
  • Line 300: “yoga is”

Author Response

Abstract and title: The phrase “mindfulness exercise” is ambiguous, as it could simply mean practicing mindfulness, and it doesn’t necessarily imply that there is a physical exercise component attached with it. The term “non-mindfulness exercise” is then even more ambiguous, as it only says that there is some form of unspecified exercise but one not involving mindfulness. I suggest adding “mindfulness” and “non-mindfulness” to the phrase “physical exercise”, to make this clearer. Alternatively, say “physical exercise involving practices of mindfulness”.

Abstract: It is not clear until the final sub-section of the abstract what the authors mean about mindfulness exercise. And it is only indirectly defined. So, yoga seems to be considered a mindfulness exercise. What are others? Again, as above, please be clearer about the terms and definitions used. Note that both Li et al. (2018) and Tsang et al. (2008) use “mindful exercise” rather than “mindfulness exercise”, which offers a further alternative.

Reply: Thank you for your comments. The wordings have been revised to “mindful” and “non-mindful” (instead of “mindfulness” and “non-mindfulness”) to reduce confusion. The wordings are now consistent with Li et al. (2018) and Tsang et al. (2008).

In Line 34, the authors state that the review focuses on anxiety disorders and not anxiety. However, in Line 35, anxiety is again used on its own. So, is anxiety used as a synonym for anxiety disorders here?

Reply: “Anxiety” and “anxiety disorders” are not synonyms in the review. This review focuses on anxiety but not anxiety disorders. P. 1, line 35 has been revised.

Line 40: grammar (“study showed”)

Reply: It was revised; please see p. 2, line 40.

The paragraph starting with Line 51 addresses my concern above. However, it is a bit rushed. Most of the literature on mindfulness relates to practices without an exercise component and thus mainly sitting meditation. This needs to be explained first before mentioning mindfulness in the context of physical exercises. Otherwise uninformed readers link the vast mindfulness literature directly to physical exercise without appreciating this distinction. The present study focuses on a relatively small part of the mindfulness literature. However, this is not a criticism but something that can be seen as a strength.

Reply: Thank you for your comments. The wordings have been revised to “mindful” and “non-mindful” (instead of “mindfulness” and “non-mindfulness”) to reduce confusion. The definition of mindful exercise is mentioned on p. 2, lines 53-54.

The introduction is a bit rushed. What did Li et al. (2018) and Tsang et al. (2008) find in their reviews? Schizophrenia is quite distinct, but anxiety is very related to depression, which means that similar results might be expected compared to Tsang et al.

Reply: Thank you for your comments. More information has been added on p. 2, lines 68-72.

Further to my point that anxiety and depression are related: Quite often, interventions measure both outcomes in their studies. So, to what extent did studies included in Tsang et al. also measure anxiety? And how many studies found in your review were also included in Tsang et al.? Any overlap is important to note – not in the introduction of course, but at some stage in the manuscript.

Reply: The included studies in this review are different from those studies included by Tsang et al.

Please mention and discuss effect sizes for these studies.

Reply: Standard mean difference (SMD) is used in this review instead of effect size. Please see below a reference that used the same approach.

Li, R., Chen, H., Feng, J., Xiao, Y., Zhang, H., Lam, C. W.-K., & Xiao, H. (2020). Effectiveness of traditional Chinese exercise for symptoms of knee osteoarthritis: A systematic review and meta-analysis of randomized controlled trials. International Journal of Environmental Research and Public Health, 17(21), 7873.

Discuss the potential of self-selection bias in the studies. Perhaps people opting into yoga interventions are somehow different to participanrts in qigong classes?

Reply: All selected studies in this review are randomized controlled trials. The participants were randomly assigned to mindful and non-mindful exercise groups. Also, studies in this review either used yoga or qigong as an intervention; no studies provided two mindful interventions for the participants to choose.

 Line 300: “yoga is”

Reply: It was revised; please see p.13, line 300 and line 304.